# Modified DNA Aptamers for C-Reactive Protein and Lactate Dehydrogenase-5 with Sub-Nanomolar Affinities

**DOI:** 10.3390/ijms21082683

**Published:** 2020-04-13

**Authors:** Hirotaka Minagawa, Yuka Kataoka, Hiroto Fujita, Masayasu Kuwahara, Katsunori Horii, Ikuo Shiratori, Iwao Waga

**Affiliations:** 1NEC Solution Innovators, Ltd., 1-18-7, Shinkiba, Koto-ku, Tokyo 136-8627, Japan; minagawa-hir@nec.com (H.M.); k-horii@nec.com (K.H.); iwa-waga@nec.com (I.W.); 2Graduate School of Integrated Basic Sciences, Nihon University, 3-25-40 Sakurajosui, Setagaya-ku, Tokyo 156-8550, Japan; kataoka.yuka@nihon-u.ac.jp (Y.K.); fujita.hiroto@nihon-u.ac.jp (H.F.)

**Keywords:** C-reactive protein, lactate dehydrogenase, aptamer, base-appended base modification

## Abstract

Human C-reactive protein (CRP) and lactate dehydrogenase are important markers in clinical laboratory testing—the former is used to detect in vivo inflammation, and the latter is used to detect cell necrosis and tissue destruction. We developed aptamers that bind to human CRP and human lactate dehydrogenase-5 (LDH-5) with high affinities (dissociation constants of 6.2 pM and 235 pM, respectively), applying the systematic evolution of ligands by exponential enrichment (SELEX) method, and by using a modified DNA library containing the following base-appended base modifications: analog adenine derivative at the fifth position of uracil (U^ad^), analog guanine derivative at the fifth position of uracil (U^gu^), and analog adenine derivative at the seventh position of adenine (A^ad^). A potential application of these aptamers as sensor elements includes high-sensitivity target detection in point-of-care testing.

## 1. Introduction

Aptamers are single-stranded DNA or RNA oligonucleotides that bind to specific molecules or cells [1,2]. Aptamers have similar antibody functions, but are easier to manufacture, and they are advantageous due to low immunogenicity, easy chemical modification, and strong thermal degeneration [3,4,5]. As such, the development of aptamers as therapeutic and diagnostic agents [6], and as molecule detection devices in biosensors [7], is of current interest.

Aptamers are separated from large nucleic acid combinatorial libraries using an iterative selection process called SELEX (systematic evolution of ligands by exponential enrichment) [8]. In brief, the nucleic acid combinatorial library consists of sequences with a primer and random regions, and the sequence that binds to the target is selected [9,10].

Alternative approaches using a modified nucleic acid library in the SELEX method can improve the probability of detecting candidate aptamers [11]. For instance, SomaLogic Inc. has developed an aptamer called SOMAmer (slow, off-rate, modified aptamer) using uridine derivatives, in which a foreign functional group, such as highly hydrophobic amino acid side chains, is added to the base [12]. Moreover, TAGCyx Biotechnologies has developed aptamers using 7-(2-thienyl)-imidazo [4–b] pyridine (DS)-2-nitro-4-propynylpyrrole (Px) base pairs [13].

In this study, we have developed an alternative method based on base-appended base (BAB) analogs, and successfully isolated aptamers capable of binding to a variety of targets with high affinity [14,15,16,17]. Human C-reactive protein (CRP) is a main acute-phase protein, and is an important in vivo inflammation marker, as the CRP concentration may increase by more than 1000-fold in a severe inflammatory state [18,19]. Saliva is a biological sample that can be collected non-invasively, and the CRP concentration in saliva was reported to be highly correlated to the CRP concentration in serum [20,21]. Despite the low concentration of CRP in saliva, which is approximately 1/1600 of the concentration in the serum [20], this non-invasive procedure is favored for assessing cardiovascular disorders and its association with oral diseases [22,23], and as a marker of pneumonia in infants [24].

Currently, the principal methods for CRP measurement are immunoturbidimetric assay and latex agglutination turbidimetry. Specific monoclonal anti-CRP antibodies are available from commercial sources, and are also used as detection elements in biosensors [25]. However, these antibody-based methods are costly [26]. Lactate dehydrogenase (LDH) is a detectable enzyme in the cytoplasm of nearly all cells in the human body, and its extracellular presence is associated with cell necrosis and tissue destruction, since it becomes extracellular due to cell death [27,28]. LDH is an enzyme whose activity is routinely measured clinically, and it has subtypes ranging from LDH-1 to LDH-5 [29]. The isoenzyme profile of LDH activity is different in the saliva and in the blood: while LDH-5 is dominant in the former, LDH-1 is dominant in the latter [30]. It has been reported that abnormalities of LDH-5 in the saliva are associated with oral squamous cell carcinoma [31,32]. Thus, LDH can serve as an early tumor marker [33]. LDH measurements are mainly based on activity measurement, and anti-LDH antibodies for research applications are available from commercial sources [34]; however, to the best of the authors’ knowledge, no diagnostic application based on LDH detection has been developed yet.

In this study, we isolated artificial nucleic acid aptamers with high affinity for CRP and LDH-5, using three types of libraries incorporating BAB modification bases (analog adenine derivative at the fifth position of uracil (U^ad^), analog guanine derivative at the fifth position of uracil (U^gu^), and analog adenine derivative at the seventh position of adenine (A^ad^)). The aptamer selection was performed utilizing a nucleic acid library with three types of modified nucleic acids, namely U^ad^, U^gu^, and A^ad^ (Figure 1), to obtain CRP-binding and LDH-5-binding aptamers.

## 2. Results

In the selection of CRP-binding aptamers, six sequences with more than 5% sequence ratio were obtained in the Round 8 pool for U^ad^, six sequences with more than 5% sequence ratio were obtained in the Round 8 pool for U^gu^, and four sequences with more than 5% sequence ratio were obtained in the Round 8 pool for A^ad^ (Table A1). In the selection of LDH-5-binding aptamers, five sequences with more than 5% sequence ratio were obtained in the Round 8 pool for U^ad^, four sequences with more than 5% sequence ratio were obtained in the Round 8 pool for U^gu^, and two sequences with more than 5% sequence ratio were obtained in the Round 8 pool for A^ad^ (Table A1).

Target binding assessment was performed using surface plasmon resonance (SPR) on candidate U^ad^-, U^gu^-_,_ and A^ad^-containing aptamers with high sequence ratios. For CRP, six out of the six candidate U^ad^ sequences and five out of the six candidate U^gu^ sequences showed target binding, whereas none of the four candidate A^ad^ sequences showed binding (Table 1). Some U^ad^ candidate sequences showed binding with secretory immunoglobulin A (SIgA) (Figure A1), a protein molecule that has no association with the sequence or structure of CRP, and thus were removed. The remaining candidates that showed no SIgA binding, namely Ugu2 and Ugu4 (Figure A2), were selected as CRP-binding aptamer candidates for the next step.

For LDH-5, three out of the three candidate U^ad^ sequences, four out of the four candidate U^gu^ sequences, and two out of the two candidate A^ad^ sequences showed target binding (Table 1). Since none of the candidate sequences showed any SIgA binding (Figure A3, Figure A4, Figure A5), U^ad1^, U^gu3^, and A^ad1^ are the best ones, because they are the ones with the highest target affinity. Therefore, U^ad1^, U^gu3^, and A^ad1^ were selected as LDH-5-binding aptamer candidates for the next step.

Aptamer-conjugated magnetic beads were produced using the two candidate sequences (U^gu2^ and U^gu4^) for CRP, and CRP pull-down assays were conducted using purified CRP dissolved in selection buffer (SB; 40 mM 4-(2-hydroxyethyl)-1-piperazineethanesulfonic acid (HEPES), pH7.5, 125 mM NaCl, 5 mM KCl, 1 mM MgCl_2_, 0.01% Tween 20) or a human saliva sample. The results showed that only U^gu4^ was able to bind CRP in the SB buffer and in the human saliva sample (Figure 2a). This shows that U^gu4^ can specifically bind to CRP even in the presence of other contaminant proteins, and thus was selected as the final CRP-binding aptamer candidate.

The same aptamer-conjugated magnetic beads approach was used for LDH-5, to test the three candidate sequences (U^ad1^, U^gu3^, and A^ad1^). The pull-down assays performed on SB buffer containing purified LDH-5 and a human saliva sample showed that all the candidate sequences were able to bind LDH-5 (Figure 2b). This shows that U^ad1^, U^gu3^, and A^ad1^ can bind specifically to LDH-5 in the presence of other contaminant proteins.

The cross-reaction between LDH-1 (LDH-5 isozyme) and the three LDH-5 candidate aptamers (U^ad1^, U^gu3^, and A^ad1^) were evaluated using SPR response measurements. In this setting, only A^ad1^ did not show any LDH-1 binding (Figure A6). The LDH-5 amino acid sequence has 75% homology with the LDH-1 amino acid sequence, and considering that A^ad1^ had very high LDH-5 binding specificity, it was selected as the final LDH-5-binding aptamer candidate.

Generally, when the aptamer sequence is long, it is more likely to form various secondary structures that destabilize the conformation of the target binding site of the aptamer [35]. Thus, minimizing the aptamer sequence is a viable strategy to suppress structural instability, thereby improving target-binding strength [36]. Accordingly, sequence minimization for CRP–U^gu4^ and LDH–A^ad1^ was conducted using the SPR measurement as an indicator. Various variant sequences were produced by truncating the 3′ and 5′ ends of the candidate sequences (Figure 3a and Figure 4a). We successfully shortened the CRP–U^gu4^ to a 48 base candidate (CRP–U^gu4-3^) (Figure 3b). CRP–U^gu4-3^ showed higher CRP-binding avidity (dissociation constant (Kd) = 6.2 pM) (Figure 3c) than the original CRP–U^gu4^ (Kd = 53.4 pM). In contrast, shortening by truncation of the 3′ end (CRP–U^gu4-4^) resulted in the loss of CRP-binding avidity, even by using the same sequence length of CRP–U^gu4-3^ (Figure 3b).

We then successfully shortened the LDH–A^ad1^ candidate to 44 bases (LDH–A^ad1-3^) (Figure 4b). LDH–A^ad1-3^ showed much higher LDH-5 binding avidity (Kd = 235 pM) (Figure 4c) than the original LDH–A^ad1^ (Kd = 1.68 nM). LDH–A^ad1-4^, where the primer region at the 3′ end was truncated, showed extremely weak-avidity target-binding (Figure 4b).

The predicted secondary structure of the truncated sequences was compared using the VALFold program [37] and general DNA parameters [38], and using sequences by setting U^gu^ as “T” and Aad as “A”. The results showed a large difference in predicted secondary structures between CRP–U^gu4^–CRP-U^gu4-3^ and CRP–U^gu4-4^ (Figure A7). Moreover, the predicted secondary structures for LDH–A^ad1^–LDH–A^ad1-3^ and LDH–A^ad1-4^ were also quite different (Figure A8). The results suggest that in CRP–U^gu4-4^ and LDH–A^ad1-4^, the truncation at the 3′ end affected the structure, explaining the loss of target-binding avidity. According to the predicted secondary structure of the truncated sequence of CRP–U^gu4-3^ (Figure A9), it could be capable of minimizing the CRP–U^gu4-3^ sequence.

## 3. Discussion

The aptamers isolated using the three types of base-appended base approach (U^ad^, U^gu^, and A^ad^; Figure 1), showed quite different characteristics. For CRP, we successfully obtained binding sequences with U^ad^ and U^gu^, whereas with A^ad^, although sequences with high sequence ratios were retrieved, we found no sequence capable of binding CRP (Table 1).

Aptamers for CRP have been previously obtained using natural bases [39,40,41] and modified bases [42], but the binding regions on the protein have not been reported. Analysis of the CRP crystal structure revealed that there is a positively charged region outside the groove at the bottom of Asp112, which is an important residue for the recognition of complement C1q by CRP [43]. The binding sites of the aptamers isolated in this study are currently unknown, but aptamers, which are negatively charged, usually bind easily to positively-charged regions. Considering this, it is likely that, in the case of A^ad^, the volume of the modified site was larger U^ad^ and U^gu^, and thus appropriate sequences that fit the groove structure could not be obtained.

On the other hand, for LDH-5, binding sequences with U^ad^, U^gu^, and A^ad^ were obtained, and all candidate sequences showed strong LDH-5-binding. Since LDH-5 has a substructure that interacts with dinucleotides like FADH and NADH, called a Rossmann-type fold [44], the binding to the modified U^ad^, U^gu^, and A^ad^ may have been easier

A previous study reported the isolation of LDH-binding aptamers [45], but their sequence and target specificity were not elucidated. The aptamers obtained in this study can distinguish between LDH isozymes, and could be potentially be used as tools for the early diagnosis of oral cancer [46].

The aptamers obtained in this study have sufficient potentials for detecting endogenous target molecules, according to their dissociation constants and high binding specificity. No detectable pull-down can be observed for non-spiked human saliva, because the concentrations of endogenous target molecules are below the limitation of Coomassie stain [23]. Thus, we spiked the saliva samples with recombinant analytes as a validation of binding, and showed specific bindings of the aptamers. Application of electrochemical sensing [47], acoustic sensing [48], or thermal sensing [49] may lead to highly sensitive target detection in such testing, as point-of-care testing [50,51].

## 4. Materials and Methods

### 4.1. Materials

Purified CRP was purchased from OriGene Technologies Inc. (Rockville, Maryland, United States), and purified LDH-5 was purchased from Meridian Life Science Inc. (Tennessee, United States). Purified LDH-1 isoenzyme was purchased from RayBiotech Life, United States. Magnetic beads for the immobilization of the target and recovery of biotinylated DNA—namely, Dynabeads MyOne Carboxylic Acid magnetic beads, and Dynabeads MyOne SA C1 magnetic beads—were purchased from Invitrogen (Carlsbad, CA, United States). KOD Dash (TOYOBO, Japan) was used for PCR and incorporating modified bases. Synthetic compounds used as primers, random pools, and aptamer clone templates were purchased from Integrated DNA Technologies MBL KK (IDT-MBL KK, Japan). Research grade materials were used for other reagents. The dU^ad^TP, dU^gu^TP, and dA^ad^TP were synthesized using previously reported methods [17,52,53].

### 4.2. SELEX

The SELEX method was performed as previously described [16]. CRP or LDH-5 was bound to Dynabeads MyOne Carboxylic Acid magnetic beads following the manufacturer’s instructions, and was washed with selection buffer (SB; 40 mM HEPES, pH7.5, 125 mM NaCl, 5 mM KCl, 1 mM MgCl_2_, 0.01% Tween 20), to produce the target beads. dsDNA incorporating U^ad^, U^gu^, or A^ad^ was produced using complementary strands with 5′-biotin modification (CRP: GATATGTCCAGCCTGTCGAATG C-N_30_-CTAAACTGATGTGCGGCGTAACC, LDH-5: GTATAGTAGCCAGCCAGCCTTAGG-N30-CATAAACGGCGAGGTGTCAATTCC), forward (Fw) primer (CRP: GGTTACGCCGCACATCAGT TTAG, LDH-5: GGAATTGACACCTCGCCGTTTATG). After binding the dsDNA to Dynabeads MyOne SA C1 magnetic beads, single strand DNA was eluted using 0.02 M NaOH and neutralized with 0.08 M HCl, to produce the ssDNA U^ad^, U^gu^, and A^ad^ libraries.

After mixing 80 pmol of the library to 250 μg of target beads for 15 min at 25 °C, and washing the beads with SB, the bead-bound ssDNA was eluted with 7 M urea. The eluted ssDNA was amplified by PCR using the Fw primer and biotin-modified reverse (Rv) primers. The amplified dsDNA was bound to Dynabeads MyOne SA C1 magnetic beads, and after elution of the Fw chain with 0.02 M NaOH, the beads were washed with SB. The ssDNA produced by this method, using the Rv chain and Fw primer, and either U^ad^-, or U^gu^-, or A^ad^ immobilized in magnetic beads, was used in the next round. After eight rounds of selection, PCR was performed using the Fw primer and non-biotin-modified Rv primer, and sequencing was performed using a GS junior sequencer (Roche, Indianapolis, United States)

### 4.3. Surface Plasmon Resonance (SPR) Assay

All SPR measurements were performed at 25 °C using the ProteON XPR360 instrument (Bio-Rad Laboratories, Inc., Hercules, United States) [16]. For the U^ad^ and U^gu^ aptamer clones, the ligand was set by appending poly-A_20_ at the 3′ end, and by hybridization of the 5′ end bound to an NLC sensor chip with biotin-modified oligo (dT_20_) [16,17]. For the A^ad^ aptamer clones, the ligand was set by appending poly-T_20_ at the 3′ end, and by hybridization of the 5′ end bound to NLC sensor chip with biotin-modified oligo (dA_20_) [53]. Either CRP or LDH-5 was used as an ananalyte, and SB was used as the running buffer. The dissociation constant was calculated by simple 1:1 biomolecular interaction model that is the most common kinetic fit model used for SPR data analysis, following the device instructions.

### 4.4. Pull-Down Assay

The pull-down assay using magnetic beads was performed by using a modified version of the method described in [17]. Clones (350 pmol) synthesized using the 5′ end biotinylated Fw primer, a non-biotin-modified template, and 3 mg of Dynabeads MyOne SA C1 magnetic beads, were mixed in SB (25 °C, 30 min). Then, the beads were washed three times with SB. Next, after eluting the template with 20 mM NaOH, the beads were washed three times with SB and then suspended in 300 µL of SB. Clone beads (250 µg) and CRP or LDH-5 (2 µg) were mixed in SB at 25 °C for 60 min. In parallel, a sample of 88% of human saliva was supplemented with 2 µg of CRP or LDH-5 at 25 °C for 60 min. The supernatant was removed, and after washing the beads with SB three times, the protein bound to the beads was agitated in a 2% SDS solution at room temperature for 10 min, to elute the synthetic compound bound to the beads. Electrophoresis was performed with the eluted sample using PAGEL C520L (ATTO, Tokyo, Japan) following the manufacturer’s instructions.

## 5. Conclusions

In this study, we obtained aptamers with an extremely high binding avidity for CRP and LDH-5, by applying selection using three types of BAB-modified DNA libraries (dU^ad^TP, dU^gu^TP, and dA^ad^TP). The best minimized CRP aptamer (CRP-U^gu4−3^, 48 mer) binds strongly to CRP (*K*_d_ = 6.2 pM), and the best minimized LDH-5 aptamer (LDH-A^ad1^-^−3^, 44 mer) also binds strongly to LDH-5 (*K*_d_ = 235 pM). To the best of our knowledge, this is the first report fully characterizing LDH-5 binding aptamers. the obtained LDH-5 aptamers have high specificity and the ability to distinguish between highly homologous isozymes. These aptamers could be useful as biosensor elements with an electrochemical base of the target analyte.

## Figures and Tables

**Figure 1 ijms-21-02683-f001:**
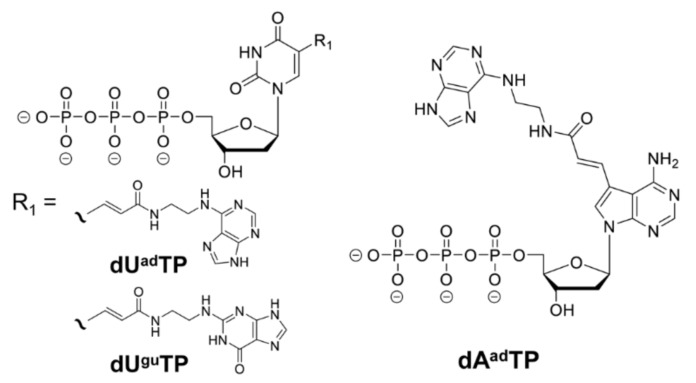
Chemical structures of dU^ad^TP, dU^gu^TP, and dA^ad^TP.

**Figure 2 ijms-21-02683-f002:**
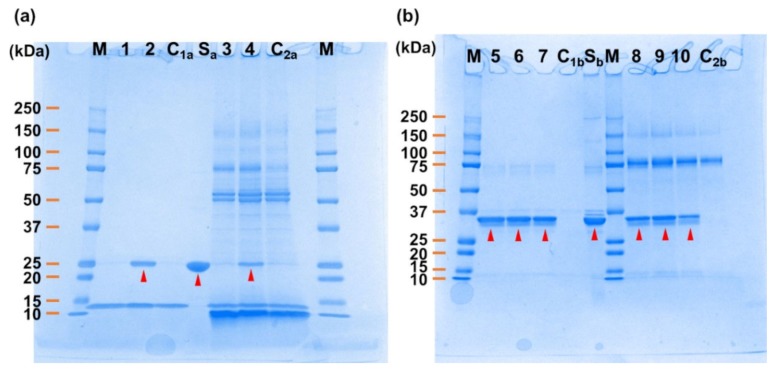
SDS-PAGE image of the samples, representative of the pull-down assay. (**a**) Pull-down assay using CRP in selection buffer (SB) or human saliva. Lane M: Precision Plus Protein Kaleido Scope Standards (Bio-Rad, United States); Lanes 1–2: CRP in SB sample eluted from the CRP–U^gu2^ and U^gu4^-immobilizing CRP capture beads, respectively; Lane C_1a_: CRP in SB buffer sample eluted from the control beads; Lane S_a_: CRP; Lanes 3–4: spiked CRP in human saliva sample eluted from the CRP– U^gu2^ and U^gu4^-immobilizing CRP capture beads, respectively; Lane C_2a_: spiked CRP in human saliva sample eluted from the control beads. (**b**) Pull-down assay using LDH-5 in SB or human saliva. Lane M: Precision Plus Protein Kaleido Scope Standards (Bio-Rad, United States); Lanes 5–7: LDH-5 in SB sample eluted from the LDH–U^ad1^, U^gu3^, and A^ad1^-immobilizing LDH-5 capture beads, respectively; Lane C_1b_: LDH-5 in SB buffer sample eluted from the control beads; Lane S_b_: LDH-5; Lanes 8–10: spiked LDH-5 in human saliva sample eluted from the LDH–U^ad1^, U^gu3^, and A^ad1^-immobilizing LDH-5 capture beads, respectively; Lane C_2b_: spiked LDH-5 in human saliva sample eluted from the control beads.

**Figure 3 ijms-21-02683-f003:**
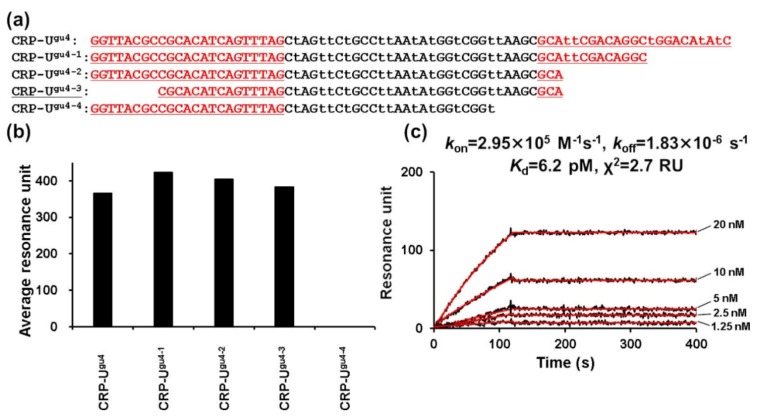
CRP–U^gu4^ truncated aptamer preparation. (**a**) Sequences of the CRP–U^gu4^ truncated aptamers. Sequences are aligned in the 5′ to 3′ direction. Underlined regions derive from the primer or primer-binding regions. The letter “t” indicates U^gu^. (**b**) SPR response units of the interaction between the CRP and the CRP–U^gu4^ truncated aptamers. Measurements were performed with multicycle kinetics, and CRP (400 nM) was injected over the respective aptamer-immobilizing sensor chips for 120 s at a flow rate of 50 μL/min. SPR response units between 115–125 s, in the plateau region of the sensorgram curves, were averaged. (**c**) Representative SPR sensorgrams showing the interaction between the CRP and the aptamer CRP–U^gu4−3^. Various concentrations of CRP (1.25–20.00 nM) were injected over the respective CRP-U^gu4−3^-immobilizing sensor chip for 120 s at a flow rate of 50 μL/min. The black line represents the measured curve, and the red line represents the fitting curve. The average of the squared differences between the measured data points and the corresponding fitted values are represented as χ^2^.

**Figure 4 ijms-21-02683-f004:**
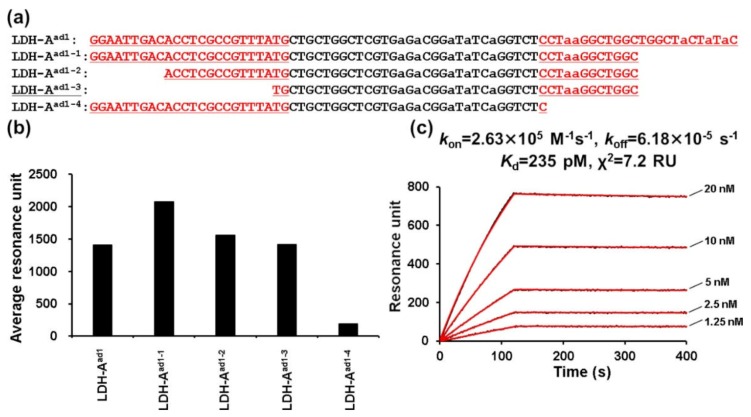
LDH–A^ad1^ truncated aptamer preparation. (**a**) Sequences of the LDH–A^ad1^ truncated aptamers. Sequences are aligned in the 5′ to 3′ direction. Underlined regions derive from the primer or primer-binding regions. The letter “a” indicates A^ad^. (**b**) SPR response units of the interaction between the LDH-5 and the LDH-A^ad1^ truncated aptamers. Measurements were performed with multicycle kinetics, and LDH (400 nM) was injected over the respective aptamer-immobilizing sensor chips for 120 s at a flow rate of 50 μL/min. SPR response units between 115–125 s, which were in the plateau region of the sensorgram curves, were averaged. (**c**) Representative SPR sensorgrams showing the interaction between the LDH-5 and the aptamer LDH–A^ad1-3^. Various concentrations of LDH-5 (1.25–20.00 nM) were injected over the respective LDH-A^ad1-3^-immobilizing sensor chip for 120 s at a flow rate of 50 μL/min. The black line represents the measured curve, and the red line represents the fitting curve. The average of the squared differences between the measured data points and the corresponding fitted values are represented as χ^2^.

**Table 1 ijms-21-02683-t001:** The sequences from the U^ad^-, U^gu^-, and A^ad^-modified single strand DNA library that target binding assessment was performed using surface plasmon resonance (SPR).

Clone Name	Sequence Ratio ^(a)^	Random Region Sequence ^(b)^	Average Resonance Unit ^(c)^ for 400 nM Target
CRP-U^ad1^	27.5%	CGGttACAGAtGAtCAGGCtCAAACAGAtt	341
CRP-U^ad2^	24.6%	AACtGGGtCGAACGCGGttACAGAtGAtCAG	427
CRP-U^ad3^	17.4%	CCttCAAGAAttGCGGttACAGAtGAtCAG	236
CRP-U^ad4^	11.9%	CCtCGtAtAAGACGGttACAGAtGAtCAGG	396
CRP-U^ad5^	10.7%	tGAtAAAAGCCCGGttACAGAtGAtCAGGG	479
CRP-U^ad6^	7.3%	ACCGGttACAGAtGAtCAGGGGCAAAGAAC	249
CRP-U^gu1^	27.9%	CAtGAAtGCGtGtGGttAtAGtAttGAACt	157
CRP-U^gu2^	14.3%	GtCtGAAAtCGCtttCCGGAtCGGACttAA	435
CRP-U^gu3^	9.6%	GACGtCCCACGGtttGAtCAAACGtACAtA	18
CRP-U^gu4^	6.5%	CtAGttCtGCCttAAtAtGGtCGGttAAGC	362
CRP-U^gu5^	5.9%	ACtCAAGttAtGCtGGACttCtttACAAAC	101
CRP-U^gu6^	5.2%	GCAtACAACtCCCtAGtCAAACtGACAttA	93
CRP-A^ad1^	39.2%	aaCaTTGaGTGCCaTGCCCTTCGTaGaCa	–53
CRP-A^ad2^	19.5%	TTTaCCGaaTGCCaTGCCCGaGaGTaGaCa	–15
CRP-A^ad3^	10.0%	TCGaaCGCCaTGCCaCTGCCCGGTTaGaCa	–13
CRP-A^ad4^	5.1%	aCGTaGCaTaGTGTaaGGaGCGCCCaCTaT	–12
LDH-U^ad1^	8.4%	CACCCtCCAGACtAtAttCtAGGCAACCGA	1583
LDH-U^ad2^	7.5%	tGtGtCGAtCAGAtGCGttACtAAAtCtCA	1441
LDH-U^ad6^	4.5%	tGGGCtAtGGtACtAGACtGGCtCGGttGC	829
LDH-U^gu1^	24.3%	CCtCCGCttGtGGAtACGAtGGACtAGtGG	1011
LDH-U^gu2^	12.0%	ACCttAGACACGGtACttACCGACACtAAA	862
LDH-U^gu3^	8.9%	ttAGAtACttGGCtCtACttAttGACAAtC	1255
LDH-U^gu4^	7.3%	CACtCCtGAttGCttAAGAtCttAGttCGA	705
LDH-A^ad1^	55.8%	CTGCTGGCTCGTGaGaCGGaTaTCaGGTCT	1415
LDH-A^ad2^	14.9%	aGaGGGaGaTCaTCTCTCTGGCGGaCaCaa	518

(^a^) Sequence ratio was defined as the ratio of the sequence from the total number of sequences that were generated by next-generation sequencing. (^b^) Letter “t” indicates the analog adenine derivative at the fifth position of uracil (U^ad^) or the analog guanine derivative at the fifth position of uracil (U^gu^), and letter “A” indicates the analog adenine derivative at the seventh position of adenine (A^ad^). (c) SPR response units of the interaction between the target and the selected clones. Measurements were performed with either human C-reactive protein (CRP) or lactate dehydrogenase (LDH)-5 (400 nM), injected over the respective aptamer-immobilizing sensor chips for 120 s at a flow rate of 50 μL/min. SPR response units between 115–125 s, which were in the plateau region of the sensorgram curves, were averaged.

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
