# Peer review of "Modified DNA Aptamers for C-Reactive Protein and Lactate Dehydrogenase-5 with Sub-Nanomolar Affinities"

_ijms, 2020, doi:10.3390/ijms21082683_

Round 1

Reviewer 1 Report

Article is well-written and the overall workflow to select best CRP- and LDH-5-aptamers is well-designed/described. Only minor points need to be addressed before publication. 1) Conclusions should be rewritten to better summarize the important findings of the whole article as well as to add findings about CRP aptamer, not mentioned at all in this section. 2) Errors on dissociation constants must be reported. 3) In lines 159-161 and 175-198 (Discussion section) authors focused on explaining why the aptamers discarded (as a result of their selection workflow) are unable of target binding. In my opinion, article can be significantly improved if, after simple inspection of VALFold results, authors add some hypotheses about structural motifs they found to be common to the BEST binding aptamers, thus finally correlating aptamer structures to high binding affinity. I guess these common motifs may guide the binding. 4) Lines 132-133: please make it clear that the best shortened aptamer is, among the ones with similar affinity as the parent aptamer, the shortest one. 5) Lines 81 and 98: specify what SIgA and SB stand for. 6) Lines 95-96: spell out that Uad1, Ugu3 and Aad1 are the best ones because they are the ones with the highest target affinity. 7) Why LDH Uad3, Uad4 and Uad5 are missing in Table 1? They are good, putative aptamers (ratio>5%). I expected those in Table 1 and not LDH Uad6 featured by a ratio lower than 5%. Clarify this point. 8) Line 85: please better explain what is in fact in Table 1 (i.e. only the sequences, selected among the ones in Table A1, with ratio higher than 5%). 9) Lines 65: add here (into introduction section) information about base modifications as well as Figure 1 (i.e. move what is written in lines 67-69 to introduction). 10) Lines 35-36: change to “Alternative approaches...can improve the probability... For instance...” 11) Lines 22-23: change to "A potential application of these aptamers as sensors elements includes high-sensitivity target detection in point-of-care testing."

Author Response

Reply to Reviewer #1

I am most grateful for the careful reading of our manuscript and for the useful comments and suggestions. Our point-by-point responses, marked in red and in brackets, have been inserted into your comments as follows.

We feel that the revised manuscript is a suitable response to the comments, and is significantly improved over the initial submission. We expect that you will find the present manuscript suitable for publication to this journal. We would be glad to respond to any further questions and comments that you may have.

Reviewer(s)’ Comments to Author:

Reviewer #1

Comments to Author:

  1. Conclusions should be rewritten to better summarize the important findings of the whole article as well as to add findings about CRP aptamer, not mentioned at all in this section.

[Thank you very much for your helpful comments and suggestions. We add findings about CRP aptamer. The relevant descriptions lines 321–325 are now appropriately placed in the revised manuscript.]

  1. Errors on dissociation constants must be reported.

[We thank you very much for your careful reading of our manuscript and for your insightful comments. We have carried out single experiment for the determination of the dissociation constants. Measurements were performed with multicycle kinetics, and different CRP or LDH-5 concentrations (1.25–20 nM) were injected over the respective aptamer-immobilizing sensor chip for 120 s at a flow rate of 50 μL/min. We showed the average of the squared differences between the measured data points and the corresponding fitted values in Figure 3, and Figure 4.]

  1. In lines 159-161 and 175-198 (Discussion section) authors focused on explaining why the aptamers discarded (as a result of their selection workflow) are unable of target binding. In my opinion, article can be significantly improved if, after simple inspection of VALFold results, authors add some hypotheses about structural motifs they found to be common to the BEST binding aptamers, thus finally correlating aptamer structures to high binding affinity. I guess these common motifs may guide the binding.

[Thank you for your helpful comments and suggestions with much knowledge and insights. We have predicted the secondary structure of the truncated sequence of CRP-Ugu4-3 using VALFold program and added the potential secondary structures data in Figure A9. The relevant descriptions (lines 241–242) are now appropriately placed in the revised manuscript.]

  1. Lines 132-133: please make it clear that the best shortened aptamer is, among the ones with similar affinity as the parent aptamer, the shortest one.

[We thank you very much for your careful reading of our manuscript and for the useful suggestions. We have revised the manuscript following your suggestions (lines 135-136 and lines 139-141).]

  1. Lines 81 and 98: specify what SIgA and SB stand for.

[We thank you very much for your careful reading of our manuscript and for the useful suggestions. We specified SIgA and SB at lines 96 and 114, respectively.]

  1. Lines 95-96: spell out that Uad1, Ugu3 and Aad1 are the best ones because they are the ones with the highest target affinity.

[We thank you very much for your careful reading of our manuscript and for the useful suggestions. We have revised the manuscript following your suggestions (lines 109-110).]

  1. Why LDH Uad3, Uad4 and Uad5 are missing in Table 1? They are good, putative aptamers (ratio>5%). I expected those in Table 1 and not LDH Uad6 featured by a ratio lower than 5%. Clarify this point.

[We thank you very much for your careful reading of our manuscript and your insightful comments. Although we were a little concerned to meet your requirement sufficiently, we could not evaluated all candidates because it was costly. The 3'region sequence of LDH Uad3, Uad4 and Uad5 were similar to LDH Uad1 than LDH Uad6, therefore we selected Uad6 for target binding assesment.]

  1. Line 85: please better explain what is in fact in Table 1 (i.e. only the sequences, selected among the ones in Table A1, with ratio higher than 5%).

[We thank you very much for your careful reading of our manuscript and for the useful suggestions. We revised the caption of Table 1 (lines 99-100).]

  1. Lines 65: add here (into introduction section) information about base modifications as well as Figure 1 (i.e. move what is written in lines 67-69 to introduction).

[We thank you very much for your careful reading of our manuscript and for the useful suggestions. We moved what is written in lines 67-69 and Figure 1 to introduction section. The relevant descriptions (lines 65–67 and Figure 1) are now appropriately placed in the revised manuscript.]

  1. Lines 35-36: change to “Alternative approaches...can improve the probability... For instance...”

[We thank you very much for your careful reading of our manuscript and for the useful suggestions. We have revised the manuscript following your suggestions (lines 35-36).]

  1. Lines 22-23: change to "A potential application of these aptamers as sensors elements includes high-sensitivity target detection in point-of-care testing."

[We thank you very much for your careful reading of our manuscript and for the useful suggestions. We have revised the manuscript following your suggestions (lines 22-23).]

Reviewer 2 Report

In this paper, the author describe novel aptamer sequences that they selected to target C-reactive protein (CRP) and lactate dehydrogenase-5 (LDH-5) using SELEX. The authors employed a previously published technique in which modified nucleotides by base-appended base modification were used. The authors identified sequences with remarkably high biding affinities to either protein as determined by the SPR measurements. The authors then validated the binding in a biological context using human saliva by pull-down assay. The authors also tested the specificity of their aptamers to LDH-5 by comparing binding affinities to LDH-1, an isozyme of LDH-5, to find that one particular aptamer candidate (LDH-Aad1) has remarkably very high binding specificity to LDH-5. Finally, the authors conducted a series of truncation experiments to both LDH-Aad1 and CRP-Ugu4 to identify LDH-Aad1-3 and CRP-Ugu4-3, significantly truncated sequences that preserve the remarkable binding affinities (235 and 6.2 pM, respectively).

In general, the experiments were well-thought, executed and presented, and the results novel and important. The manuscript was also concise and well-written. I have only one comment regarding the authors’ claim that these aptamers could be used as “sensor elements with high specificity target detection in point-of-care detection” (line 22). In fact, the authors did not show in any way a feasible detection of biological CRP or LDH-5, even though they have marginally used saliva samples. Indeed, the statement line 104: “The pull-down assays performed on SB buffer containing purified LDH-5 and a human saliva sample showed that all the candidate sequences were able to bind LDH-5 (Fig. 2b)” is misleading, because it hints as if the biological saliva LDH-5 was detected, but the fact is that the saliva was spiked with exogenous analyte (line 247) negates this conclusion. I would have thus liked to see a pull-down assay without spiking with exogenous analyte –for both LDH-5 and CRP and with using primarily LDH-Aad1-3 and CRP-Ugu4-3, since these were the authors’ finalist candidates. Idem, I would have also liked to see the specificity experiment of LDH-Aad1 being tested in a real biological saliva sample as a proof that these aptamers could be indeed developed into real-time biosensors. Since the authors noted in the introduction that LDH-5 dominate in saliva whereas LDH-1 dominate in serum, they can also use serum samples for validation. This also goes for CRP in order to assess the feasibility of the concept in general. Otherwise, these aptamers could not claimed useful as non-invasive biosensors despite their remarkable binding affinities and the authors’ conclusion is a stretch of their results.

Author Response

Reply to Reviewer #2

I am most grateful for the careful reading of our manuscript and for the useful comments and suggestions. Our point-by-point responses, marked in red and in brackets, have been inserted into your comments as follows.

We feel that the revised manuscript is a suitable response to the comments, and is significantly improved over the initial submission. We expect that you will find the present manuscript suitable for publication to this journal. We would be glad to respond to any further questions and comments that you may have.

Reviewer(s)’ Comments to Author:

Reviewer #2

I have only one comment regarding the authors’ claim that these aptamers could be used as “sensor elements with high specificity target detection in point-of-care detection” (line 22). In fact, the authors did not show in any way a feasible detection of biological CRP or LDH-5, even though they have marginally used saliva samples. Indeed, the statement line 104: “The pull-down assays performed on SB buffer containing purified LDH-5 and a human saliva sample showed that all the candidate sequences were able to bind LDH-5 (Fig. 2b)” is misleading, because it hints as if the biological saliva LDH-5 was detected, but the fact is that the saliva was spiked with exogenous analyte (line 247) negates this conclusion. I would have thus liked to see a pull-down assay without spiking with exogenous analyte –for both LDH-5 and CRP and with using primarily LDH-Aad1-3 and CRP-Ugu4-3, since these were the authors’ finalist candidates. Idem, I would have also liked to see the specificity experiment of LDH-Aad1 being tested in a real biological saliva sample as a proof that these aptamers could be indeed developed into real-time biosensors. Since the authors noted in the introduction that LDH-5 dominate in saliva whereas LDH-1 dominate in serum, they can also use serum samples for validation. This also goes for CRP in order to assess the feasibility of the concept in general. Otherwise, these aptamers could not claimed useful as non-invasive biosensors despite their remarkable binding affinities and the authors’ conclusion is a stretch of their results.

[Thank you very much for your insightful comments and helpful suggestions. Although we were a little concerned to meet your requirement sufficiently, we have revised the manuscript following your suggestions at lines 22-23, lines 167-168, lines 173-174, and line 328.]

Reviewer 3 Report

This communication provides relevant information on an alternative method based on base-appended bases (BABs) showing improved target affinity. It will be of interest for the biosensing community and deserves to be published. However, details on the data analysis are not included and make the paper too specific and less reachable for a non-SPR biosensor community.

Some minor comments that might make the paper more clear:

Remark 1: In the introduction the authors do not mention any analytical techniques used in aptamer sensing, not even the one they employ, whose use they should motivate. It will be interesting for the community to cite label-free approaches such as acoustic sensing:

Peeters et al., Real time monitoring of aptamer functionalization and detection of Ara H1 by electrochemical impedance spectroscopy and dissipation mode quartz-crystal microbalance, Journal of Biosensors and Bioelectronics 5, 1000155 (2014)

and thermal sensing:

van Grinsven et al., The heat transfer method (HTM): a versatile low-cost, label-free, fast and user-friendly read-out platform for biosensor applications, ACS Applied Materials and Interfaces 6, 13309 (2014).

Remark 2: The authors should define the binding constants and avidities introduced in the graphs, as well as introduce and briefly explain the equations used to fit the data.

Remark 3: The authors should label the concentrations in the figures and should better explain the correlation between the target concentrations introduced and the binding avidities obtained.

Author Response

Reply to Reviewer #3

I am most grateful for the careful reading of our manuscript and for the useful comments and suggestions. Our point-by-point responses, marked in red and in brackets, have been inserted into your comments as follows.

We feel that the revised manuscript is a suitable response to the comments, and is significantly improved over the initial submission. We expect that you will find the present manuscript suitable for publication to this journal. We would be glad to respond to any further questions and comments that you may have.

Reviewer(s)’ Comments to Author:

Reviewer #3

  1. Remark 1: In the introduction the authors do not mention any analytical techniques used in aptamer sensing, not even the one they employ, whose use they should motivate. It will be interesting for the community to cite label-free approaches such as acoustic sensing:

Peeters et al., Real time monitoring of aptamer functionalization and detection of Ara H1 by electrochemical impedance spectroscopy and dissipation mode quartz-crystal microbalance, Journal of Biosensors and Bioelectronics 5, 1000155 (2014)

and thermal sensing:

van Grinsven et al., The heat transfer method (HTM): a versatile low-cost, label-free, fast and user-friendly read-out platform for biosensor applications, ACS Applied Materials and Interfaces 6, 13309 (2014).

[We thank you very much for your valued comments. We added references of acoustic sensing and thermal sensing technique (Ref. 48 and 49, respectively). The relevant descriptions (lines 264–267) are now appropriately placed in the revised manuscript.]

  1. Remark 2: The authors should define the binding constants and avidities introduced in the graphs, as well as introduce and briefly explain the equations used to fit the data.

[Thank you very much for your insightful comments and helpful suggestions. Although we were a little concerned to meet your requirement sufficiently, we added the description of the dissociation constant calculation. The relevant descriptions (lines 306–307) are now appropriately placed in the revised manuscript.]

  1. Remark 3: The authors should label the concentrations in the figures and should better explain the correlation between the target concentrations introduced and the binding avidities obtained.

[We thank you very much for your careful reading of our manuscript and for the useful suggestions. We labeled the target concentrations in Figure 3, and Figure 4.]

Round 2

Reviewer 2 Report

In their revised version, the authors chose the easier route to tune down their claim instead of correcting the flaw in the experimental design. I still would like to see these novel aptamers applied to authentic biological samples, maybe in the next manuscript. But for now, the authors, in their favor, need to further discuss the results: (1) arguing why they chose to spike their biological sample (i.e., tried without spiking but no detectable pull-down was observed?); (2) If that is the case, explain that this lack of real sample detection was not discouraging because the limitation rather lays in the detection method (coomassie stain); and the authors, (3) as a proof of concept/validation of binding, spiked human saliva with recombinant analytes.

Author Response

Reply to Reviewer #2

I am most grateful for the careful reading of our manuscript and for the useful comments and suggestions. Our point-by-point responses, marked in red and in brackets, have been inserted into your comments as follows.

We feel that the revised manuscript is a suitable response to the comments, and is significantly improved over the previous version. We expect that you will find the present manuscript suitable for publication to this journal. We would be glad to respond to any further questions and comments that you may have.

Reviewer(s)’ Comments to Author:

Reviewer #2

In their revised version, the authors chose the easier route to tune down their claim instead of correcting the flaw in the experimental design. I still would like to see these novel aptamers applied to authentic biological samples, maybe in the next manuscript. But for now, the authors, in their favor, need to further discuss the results: (1) arguing why they chose to spike their biological sample (i.e., tried without spiking but no detectable pull-down was observed?); (2) If that is the case, explain that this lack of real sample detection was not discouraging because the limitation rather lays in the detection method (coomassie stain); and the authors, (3) as a proof of concept/validation of binding, spiked human saliva with recombinant analytes.

[Thank you for your helpful comments and suggestions with much knowledge and insights. Although we were a little concerned to meet your requirement sufficiently, we have revised the manuscript following your suggestions at lines 264-269.]
